# Data Analysis and System Development for Medical Professionals on Sleep Apnea Syndrome and Orthostatic Dysregulation by Processing-Healthcare Professionals and Patients

**DOI:** 10.3390/s22031254

**Published:** 2022-02-07

**Authors:** Miyori Shirasuna

**Affiliations:** National Institute of Technology, Tsuruoka College, Yamagata 997-8511, Japan; m-shirasuna@tsuruoka-nct.ac.jp

**Keywords:** medical engineering, medical assistance, multivariate analysis, sleep apnea syndrome, orthostatic regulation disorder

## Abstract

This paper presents the details of our research and the activities involved. Japan is one of the most advanced countries in medicine worldwide. However, in terms of technology, knowledge sharing, and successor development, Japanese medicine lags behind other developed countries, and these matters require addressing. The country is also facing a shortage of doctors, among other things, and this medical problem will surely become critical in the near future. In this study, we aim to help solve such problems from the medical engineering viewpoint, analyze and create systems based on the experience of doctors from the engineering viewpoint, and make it easy for patients to understand orthodox and general statistical analysis methods. We perform a visualization and quantitative medical data analysis and examine diagnostic support. We consider sleep apnea syndrome (SAS), and orthostatic dysregulation (OD) in children in this study. This research aims to detect SAS early, identify people with pre-SAS who are likely to become SAS in the near future, and identify OD. We analyze and identify these diseases through statistics and a multivariate analysis and create a dedicated analysis system for them. Our research and system development will allow specialists to make informed diagnoses, reproduce empirical rules, improve work efficiency, and improve patients’ health awareness. This research has only looks at two diseases, but these methods can be expected to be applied to other diseases.

## 1. Introduction

Although Japan is at the forefront of medical technology worldwide, it has many problems and lags behind other developed countries [1,2]. There are many medical support systems in engineering. For example, systems for medical professionals include cancer diagnosis, brain infarction diagnosis, and community medical support [3,4,5,6]. Additionally, for non-medical personnel, there are health management systems using smart watches, AI-based disease identification applications, and applications for diagnosing the possibility of diabetes [7,8,9]. Some challenges currently faced by the country are declining birthrate, aging population, shortage of doctors, lack of accumulation and sharing of medical expertise, and immaturity of training successor doctors. These problems will become even more critical in the future.

To help address these problems, we are collaborating with medical professionals to examine medical diagnostic support from the biomedical engineering standpoint. The methods we used are orthodox and popular statistical analysis and time-frequency analysis, such as data mining and wavelet transformation. Our study focuses on sleep apnea syndrome (SAS) and orthostatic dysregulation (OD) in children.

The possible diagnosis of SAS, including in the reserve group, was performed only through voice. In current SAS diagnostics, patients are diagnosed based on a device attached to the body, which is stressful for them and makes accurate measurement difficult, but the voice is recorded and analyzed with an integrated circuit (IC) recorder. This method causes less stress on the patient’s mind and body. Based on data of blood pressure and heart rate, OD reproduces the subclasses that doctors use as a rule of thumb. This is because many off-the-shelf data analysis systems and tools are often difficult to maintain by a diagnostician or analyst who uses them [10,11]. This classification is important because there are several types of subclasses and the treatment methods for patients differ depending on the type, but this is largely due to the experience of doctors. Thus, we collect data analysis using a simple and easy to understand system, and maintenance and version upgrades are easy owing to its simplicity. The system presents the ability for an analysis of quantitative results about diagnosis results and rules of thumb. Because raw and standardized data are stored in a database, the system performs the analysis and obtains quantitative results. Hence, the analysis results are accurate and consistent. In addition, the data of patients and those who had no health problems used in this study were fully explained by specialists to the subjects or their parents, and the contents of the study were understood. Additionally, the personal information is provided to us in an anonymous state.

In addition, our study considers and proposes methods to present easy-to-understand diagnostic results to patients. We analyze and reproduce the experience of specialist doctors, as the visual and quantitative realization of diagnostic results is not yet popular in the medicine field of Japan. For example, SAS diagnostic results are shown only in letters and numbers, and OD results are too complex for elementary to high school patients. Therefore, solutions to the medical problems mentioned previously should be determined.

The remainder of this paper is organized as follows. Section 2 describes our approach to medical data analysis. Section 3 describes our collaboration with medical specialists. Section 4 describes the SAS, and in Section 5 OD research cases. Section 6 describes the medical support system in development. Finally, Section 7 provides a summary of this study.

## 2. Medical Data Analysis

Recently, medical data analysis has undergone rapid development, which can be attributed to the advancement of machine learning and artificial intelligence (AI) technology. For example, since the onset the coronavirus disease 2019, which continues to infect people worldwide, the daily elucidation of virus details and simulations of its infection status have been reported [12,13]. Further, biomedical engineering has been applied to the elucidation and analayse many diseases. For example, new medications, artificial organs, limb and dental implant construction have been developed.

For biomedical engineering, a data analysis is performed on time series, audio, images, other data types, or a combination thereof. Related studies are progressing in advanced medical countries; nevertheless, this remains an urgent requirement in Japan in the future. Therefore, this field should be developed and disseminated in Japan.

However, using machine learning and AI in these studies involves some problems. First, these methods require large data and cannot be applied to data with a small number of samples. Second, even if the accuracy of the analysis results is acceptable, many points should be clarified because the analysis process of these methods is a “black box”. Third, when these methods are used, the analysis accuracy is highest during delivery (during initial analysis/creation), but this accuracy declines as time passes. To maintain the accuracy of the analysis, analysts need to perform continuous maintenance or upgrades [14,15]. Fourth, many of these analytical methods are difficult to understand because understanding analysis requires a good knowledge of statistics and mathematics. Therefore, the results of the analysis cannot be easily and accurately understood by everyone, aside from engineers who are familiar with them [16]. Fifth, machine learning and AI-based data analyses, such as AI-based automatic blood pressure monitors that have been reported to be medically unreliable, leading to credibility issues [17].

The medical data that we consider in this study do not contain the required amount of data for machine learning and AI, and such cases are common in medical data. Therefore, we must analyze a limited number of data, and the ideal analysis of such cases is a “white open” analysis, which is a synonym of “black box”.

## 3. Cooperation between Doctors and Data Scientists

Doctors’ expertise, experience, and intuition are invaluable, and their diagnoses are difficult to automate. Thus, we collaborated with doctors who specialize in SAS and OD and those who are familiar with medical data analysis. During our meetings and interviews with them, third-party doctors made unclear points regarding the study results reported, medical equipment used, and analysis methods used for medical data analysis.

For example, the data of medical devices used in studies are often unavailable to users, including doctors. Even if these data are acquired, expensive software is often needed for data export. Therefore, many physicians, especially those who are not involved in large projects or research groups, lack the tools, data, and time needed to acquire and analyze certain data. Typically, they also find it difficult to analyze the data using such equipment. This is a common problem for small- and medium-sized medical institutions that are privately owned by doctors in Japan.

There are often significant gaps between clients and analysts seeking analysis sup-port. This is because analysts tend to try new or complex analytical techniques, and clients who are presented with the results obtained from these analytical methods often do not understand the content. This problem is also related to machine learning and medical-data analysis that uses AI. Although these analytical methods are convenient, they it is difficult to visualize the analytical process using them because these analysis processes are black boxes in many cases. Therefore, our study focuses on how our clients, both professional physicians and analysts, can understand, share, and discuss the analytical methods and implications of the results.

## 4. Sleep Apnea Syndrome (SAS) Research

### 4.1. About SAS

SAS is a lifestyle-related disease that has attracted scholarly attention worldwide [18]. For example, traffic accidents caused by sleep apnea in patients with SAS are occasionally reported. Many SAS patients are not treated because they have few subjective symptoms and do not have a sense of crisis, and are known as “hidden SAS patients”. It is considered that SAS patients make up a few percent of the Japanese population. In addition, SAS is a common disease in middle-aged and elderly people; nonetheless, the number of patients is increasing in the younger generation. Therefore, a simple and accurate diagnosis of SAS is required. However, there are many problems associated with SAS treatment. In Japan, as diagnosing SAS is time consuming because of the following procedure:The patient will be interviewed by a doctor. If there is a high possibility of SAS, the patient rents a device that can measure SAS at home.The patient brings the device used for the SAS measurement at home to the hospital.The patient will again go back to the hospital for the results in step (2), and if the results suggest SAS, the patient reserves a polysomnogram (PSG) for overnight laboratory admission.The patient receives PSG.The patient will return to the hospital for the results of step (4) and is treated if SAS exists.

This process can take a month or longer, and the patient may have to be out of work for an SAS diagnosis, which is difficult considering the Japanese working style. Therefore, even with an SAS diagnosis, many patients find this process grueling and abandon the SAS diagnosis or discontinue treatment prematurely. This is also a problem for medical institutions. In addition, the current SAS diagnostics are inefficient. Sleep medicine professionals spend approximately 6 h per subject to manually and visually analyze more than 10 types of data, such as blood pressure and pulse, from PSG, which is accompanied by an overnight hospital stay [19,20]. This manual and visual analysis also has problems in that the diagnostic results are inconsistent.

### 4.2. Our SAS Research

#### 4.2.1. Our SAS Diagnosis

We evaluated the possibility of SAS using only sleep apnea sound data, and the PSG did not obtain breath-sound data during sleep. This is because the data for sleep breath sounds are long and noisy, and breath sounds during 6–8 h of sleep might be inconsistent. In addition, the duration of sleep is as long as 6–8 h, and thus, retaining and analyzing these data are challenging. The World Health Organization (WHO) has established criteria for SAS diagnosis; however, these criteria do not include breath sounds during sleep. Sleep sounds of patients with SAS are characteristic. For example, the sleep sound of SAS is greater than for a person without health issues, the sound when resuming breathing from apnea is very loud, and the breathing intervals are not uniform [21,22]. Considering the SAS patients under diagnosis, there were many cases in which the possibility of SAS was identified by family members and acquaintances who noticed abnormalities in their sleep apnea sounds, which prompted them to go to the hospital. Therefore, many studies have been conducted on the possibility of diagnosing SAS using breath sounds during sleep. Recently, studies using machine learning and AI have been reported [23,24].

Our study for determining the potential of SAS using sleep apnea sounds was simple. Because we wanted to make the SAS diagnosis processes easier, as identified in the previous section, we developed a method that would allow subjects to perform SAS diagnosis at home using a simple IC recorder for the benefit of both the subject and the analyst. Because the analysis is simple, the analysis time can be shortened. In addition, this method addresses the potential problem of difficulty sleeping in a non-home sleeping environment without the device being worn on the subject’s body.

Specifically, our study record breath sounds during sleep with an IC recorder, determines and analyzes the characteristics of SAS sound, and determines the probability of SAS using a simple method of multivariate analysis. From the characteristics of SAS sounds, 15 s centered on breath sounds during sleep with high sound pressure were cut out, and five sounds data were created for each person. Additionally, there were a total of 42 subjects, 37 SAS patients, 5 healthy individuals, and adult men and women, and there was no missing data. Therefore, 210 sounds data were created and we included a flag to determine whether their data suggests them to be an SAS patient or Non SAS. Then, we cluster-analyzed these sound data from 4 to 7 by the k-means method Euclidean distance. The results for six clusters presented a silhouette coefficient of 0.81 at the highest, followed by 0.69 with five clusters. In addition, we showed that SAS patients contained one of the three healthy data clustered into these six, and patients without health problems did not contain these three data. With this method, the probability diagnosis of SAS was 100%. Additionally, the results were promising, and the diagnosis time can be reduced to about 1/40 per person [22].

In the current study, we diverted and developed analytical methods and new devices that combine the above methods with signal processing methods.

Figure 1 shows the diagnostic procedure for SAS.

#### 4.2.2. SAS Sound Visualization

Figure 2 shows an example of visualizing the respiratory state of breathing sounds during the sleep of an SAS-affected person. The presence or absence of breath sounds during sleep recorded on the IC recorder was binarized with 1 and 0 for breathing and no breathing every second, respectively. Thereafter, the values were added to the time series of each second, and the added value of each second was graphed. The section between the red dashed lines is the WHO-defined apnea state for 8 s or longer [25]. Therefore, if the slope of this graph is 0, apnea occurs. This graphing method is simple and is easier for the affected individuals to understand than the diagnostic results reported in medical terminology and numbers.

In addition, current SAS treatments often do not clearly indicate changes in pathological conditions either visually or quantitatively. Therefore, both doctors and patients cannot clearly and quantitatively understand how the condition has improved.

In another example, we used a continuous wavelet transform (CWT) to visualize breath sounds during sleep and to communicate the condition to the patient. CWT is a useful analysis method (considering time and frequency) and is expressed by Equation (1) [23,24].
(1)$$W\left(\alpha ,\;\beta \right)=\int_{-\infty }^{\infty }\frac{1}{\sqrt{\alpha }}\bar{\psi \left(\frac{t-\beta }{\alpha }\right)}f\left(t\right)dt$$

Here, α, β, t ∈ R (*R* is the real number), α>0, β ≥ 0, where 1/α represents frequency, and β represents time. The value of *W* (α, β) obtained from this equation is known as the wavelet coefficient. $\psi \;\left(t\right)$ in Equation (1) is the mother wavelet, which is the core of the decomposition of the CWT. The Gabor wavelet (GW), which is one of the Mother Wavelet (MW)s, is expressed by Equation (2) [26,27,28].
(2)$$\psi^{G}\left(t\right)=\frac{1}{\pi^{1/4}\sqrt{\delta }}e^{-\frac{t^{2}}{2{\left(\delta \right)}^{2}\;}}e^{i2\pi }$$

Here, t, δ∈R and δ>0 in Equation (2). The GW is the most popular mother wavelet because it has the highest resolution among the many proposed MWs and is compatible with human vision. We improved the GW to create an SAS-only MW. The equation is given by Equation (3).
(3)$$\psi^{SMS}\left(t\right)=\frac{1}{\pi^{\frac{1}{4}}\sqrt{\delta_{D}}\left(\sqrt{A}+\sqrt{B}\right)}\left\{e^{-\frac{{\left(At\right)}^{2}}{2{\left(\delta_{D}\right)}^{2}\;\;}}+e^{-\frac{{\left(Bt\right)}^{2}}{2{\left(\delta_{D}\right)}^{2}\;\;}}\right\}e^{i2\pi }$$

Considering Equation (3), $t,\;\delta_{D},A,B\in R$, and $\delta_{D},A,B>0$. This equation is known in this paper as the Designed MW for SAS, and we defined it as DMS. An outline of this function is presented in Figure 3. DMS is a combination of two GWs and is looser and longer than GWs.

Figure 4 is a visualization of SAS sleep and non-SAS sleep breath sounds with GW at the top, and WC calculated from the DMS at the bottom. This is the waveform of the analyzed breath sounds. Each parameter in this figure is δ=1.0 in Equation (2), and A=1.0, B=0.5 and $\delta_{D}=0.5$ in Equation (3). Figure 4 shows that the warmer the color, the stronger the unique characteristics of the waveform. Compared to GW, DMS has more concentrated warm colors and shows a more focused reaction. In addition, the DMS has a weaker non-SAS response than the GW. Because SAS is a disease with few subjective symptoms, it is useful to visually indicate the condition of the patient as described above.

## 5. Orthostatic Dysregulation (OD) Analysis

### 5.1. About OD

Approximately 700,000 elementary to high school students in Japan have OD disease, and the incidence in this age group is approximately 7% [29]. Orthostatic dysregulation is a disorder of the autonomic nervous system, specifically a disorder in which blood pressure and pulse are not well regulated. The main symptoms of OD include difficulty waking up in the morning, headache, nausea, and lack of motivation. Severe symptoms can lead to school refusal and can cause multiple social problems [29,30]. In addition, because such symptoms are not felt seriously by parents and school teachers, children with OD cannot be diagnosed, and they may be warned that they have “lazy disease”. However, OD can be recovered with proper treatment and requires early diagnosis, support, and treatment.

Recently, OD has become well known in Japan, but with few specialist doctors. Therefore, many potentially affected children are diagnosed in internal medicine facilities near their homes. When we interviewed physicians and pediatricians, they indicated that OD was difficult to diagnose. Therefore, we aim to propose a simple method for diagnosing the possibility of OD. Our suggestion facilitates a simple diagnosis before seeing a doctor specialized in OD.

### 5.2. Possible Diagnosis of OD

For the diagnosis of OD, the patient performed a new standing test in the presence of a specialist doctor. The subject stood up on his own from the bed-rest state and measured his/her blood pressure and heart rate. Because OD is a disease of the autonomic nervous system in which blood pressure and heart rate are not well linked, blood pressure and heart rate are not well linked in this test as well. Conversely, for those who have no health problems, blood pressure and heart rate are linked to simple exercise scenes in daily life. The Fisher’s correlation coefficient between the two is 0.59, and the variance is ±0.11 [31].

Based on the data from the new standing test of patients with OD, the correlation coefficient between blood pressure and heart rate was obtained for 3 m at 3 s intervals during bed rest and standing, as shown on the positioning map in Figure 5 [32].

Figure 5 shows the correlation coefficient between blood pressure and heart rate during bed rest and standing in a new standing test of 35 males and females, aged 10–18 who were patients with OD, and there was no missing data. The circles in the graph represent the positions of the correlation coefficients of OD patients. The two dashed lines in each of the vertical and horizontal directions shown by the broken lines in this map have a Fisher’s correlation coefficient of 0.59 ± 0.11. They are lightly colored squares in between the two straight lines on the vertical and horizontal axes. If the blue circle mark position is located near to the red square area, the subject is considered to have no health problem. Then, the patient closest to the red square area is indicated by an arrow. The perpendicular distance between this patient and the square is plus 0.013. The position of the correlation of this OD patient is located outside this square. Since the data of people with no health problems are taken from the paper, it is not possible to set the error range and boundaries with OD patients, but in the future, we will collect data of people with no health problems and analyze them.

Considering this map, OD-affected persons have large individual differences in the correlation coefficient in both the bed rest and the standing state, and none of them correspond to those who have no health problems. Because people with OD often have low blood pressure, the possibility of OD can be easily diagnosed based on both the blood pressure rate and the positioning map. In addition, similar to the case of SAS in Figure 4, if this map is created for each treatment, the medical condition of the affected person can be confirmed, and it becomes easier for both the doctor and the patient to understand the situation.

## 6. Medical Support System

### 6.1. About the Medical Support System

This section describes the medical support system that we are developing. The process of analysis is important in the system creation.

The Faculty of Engineering is good at using mathematical formulas as tools, but tends to depend on them. For example, engineering students can effortlessly use a library of data analysis software, but they do not have sufficient control or understanding of the functional meaning of the library and the interpretation of the results. This “file/folder thinking”, which is an image of binding knowledge and formulas into a file and stored in a folder of fields and subjects, is a feature common to Japanese engineering students.

We aim to develop a system that can provide medical support. For example, when doctors and medical professionals seek to clarify various hypotheses and thoughts in the process of analyzing patient measurement data, if the situation can be visualized during their analysis, this will be a strong support for them.

### 6.2. Medical Support System and Functional Demand

Currently, medical-support research uses machine learning and AI in many cases. However, these cannot be applied to data with a small number of samples, such as those that we have considered in this study. In addition, because the analysis process of these methods is a “black box”, the examination and verification of outliers occur after the analysis, and the analyst cannot observe the calculation process. When these methods are used, the accuracy of the analysis is the highest during the initial analysis and creation, but this accuracy declines as time passes [14,15]. Moreover, many of these analytical methods are difficult to understand, and it is not easy for anyone, other than a technician who is familiar with them, to accurately understand the results of the analysis [16].

### 6.3. Our Medical Support System

Considering some cases, the proposed system does not use the development language library; hence, the calculation process of the analysis can be clarified. This creation process has two merits. First, the system creator’s understanding of the analysis method is deepened. Second, cases exist where new ideas are born from a deeper understanding of the analysis method. The MW shown in Figure 3 is one such example, which is our medical system. Our collaborative doctor is interested in analyzing the data he holds. OD specialists make OD diagnoses based on the guidelines advocated by The Japanese Society of Psychosomatic Pediatrics, but many patients fall into exceptions and vague boundaries. Therefore, the visualization and verification of this boundary is required.

We actively exchanged opinions with doctors specializing in collaborative studies and researchers in other fields and discussed the demands of specialist doctors, objective opinions, and ideas that can be applied in other fields.

Our collaborating doctors are interested in a frequency analysis of biological signals and wish to visualize them simply by the FFT, so we are also investigating FFT methods. Figure 6 is an example of the verification how Overlap Processing (OP) is performed when the fast Fourier transform (FFT) is multiplied by the window function [33,34]. The purpose of the window function is to perform the FFT in a finite period. When a function is multiplied by a window function, a section of the function is cut out. Therefore, when the function multiplied by the window function is had FFT, FFT for a finite period can be performed. In this example, the continuity of the sine waveform of the test signal is lost, owing to the correction of the window function. Therefore, the effect of the inverse Fourier transform on the test waveform is verified.

## 7. Conclusions

In this paper, we briefly described our research approach to medical data analysis in collaboration with specialist doctors, two practical examples of activities, the creation of medical support systems, and their important features. Regarding research on SAS, we showed the possibility of diagnosis by incorporating our proposal and described how to visually present it to patients. We also proposed a simple diagnosis of OD for research on OD. We provided some examples of the development of medical support systems.

Advancement in machine learning and AI enhances daily medical and biomedical engineering techniques that are useful for data analysis. Meanwhile, only a few studies were conducted on visualization and quantification of medical treatment, expert experience, and human senses, and we hope that this topic will attract scholarly attention. Additionally, we would like to study the method used in this paper to obtain more accurate results by conducting an analysis with different parameters from those used in this paper. We will continue to follow up on what has been described in this study and collaborate with doctors and experts in various fields to help improve healthcare in Japan and across the world. We will continue our research with a view of applying SAS research to sudden death in newborns and OD research to aging hypotension.

## Figures and Tables

**Figure 1 sensors-22-01254-f001:**
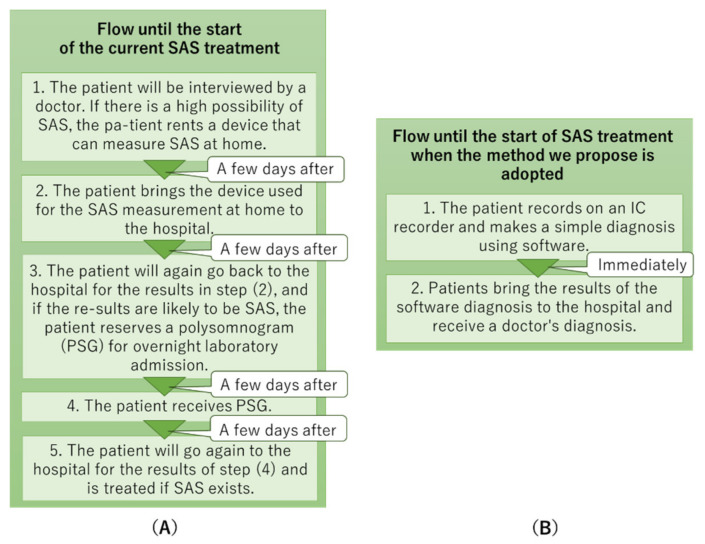
Step comparison of SAS diagnosis: (**A**) current medical institutions and (**B**) proposed procedure.

**Figure 2 sensors-22-01254-f002:**
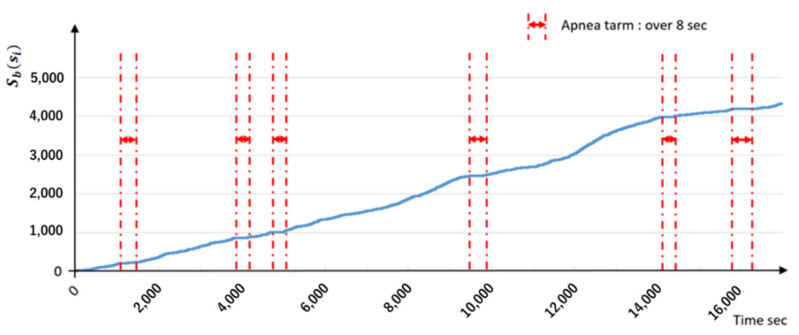
Example of visualizing the respiratory state of breathing sound during sleep of an SAS-affected person.

**Figure 3 sensors-22-01254-f003:**
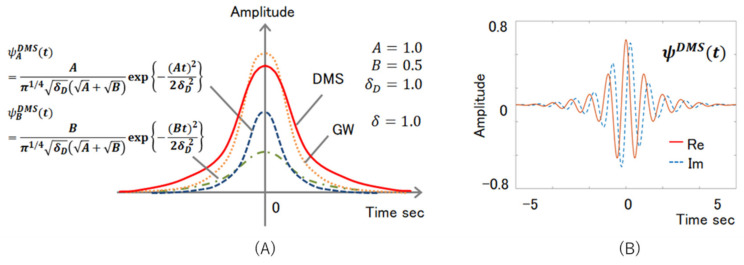
(**A**) is example of the DMS function outline. And (**B**) is the example of the DMS function external shape, the solid red line is a real number, and the broken blue line is an imaginary number.

**Figure 4 sensors-22-01254-f004:**
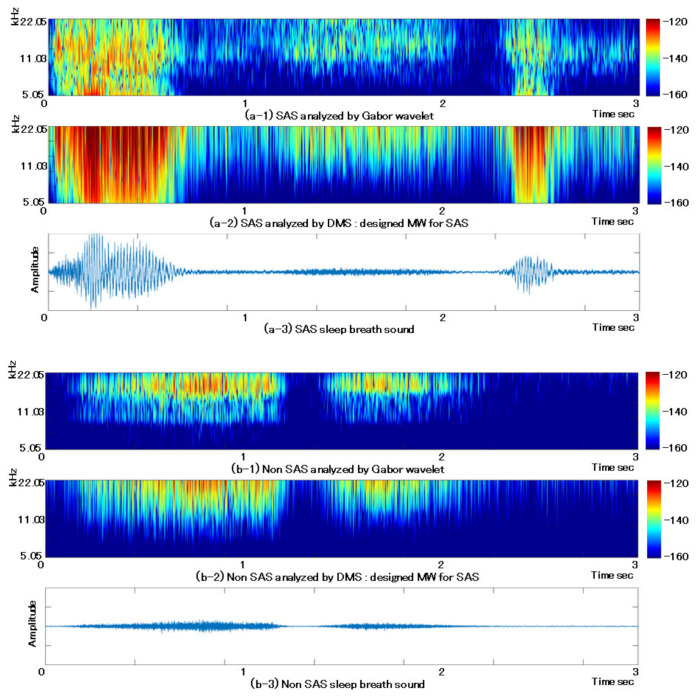
Comparison of breath sounds during sleep between SAS and non-SAS using GW and DMS.

**Figure 5 sensors-22-01254-f005:**
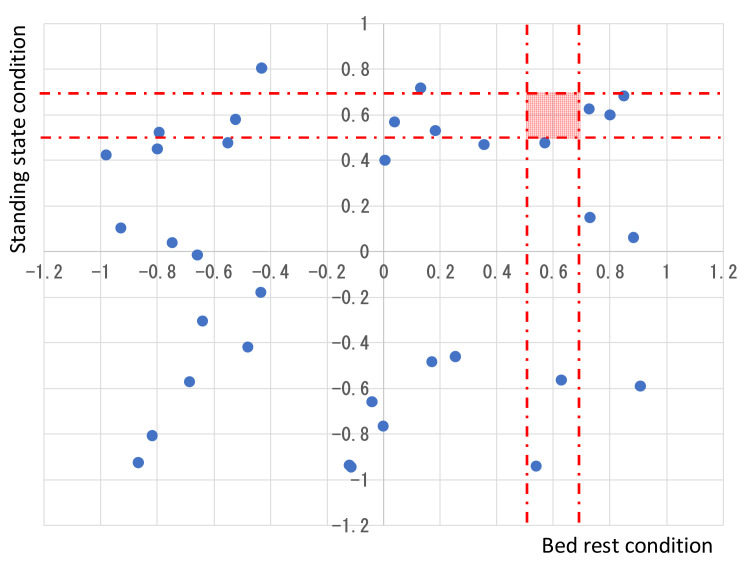
Possibility of OD determined by obtaining Fisher’s correlation coefficient of the bed rest-standing state using a positioning map.

**Figure 6 sensors-22-01254-f006:**
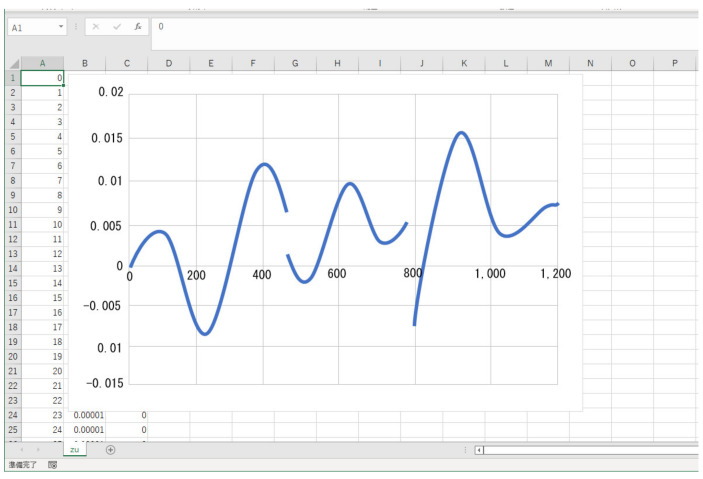
Example of our system.

## Data Availability

The data presented in this study are available on request from the corresponding author. The data are not publicly available due to the subjects who cooperated in this study provided them on the premise that the data will not be published.

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
