# Peer review of "Data Analysis and System Development for Medical Professionals on Sleep Apnea Syndrome and Orthostatic Dysregulation by Processing-Healthcare Professionals and Patients"

_sensors, 2022, doi:10.3390/s22031254_

Round 1

Reviewer 1 Report

This paper presents two visualization examples and one medical support system based on a device to diagnose SAS and OD in children cohorts. The goal is to provide a system providing data analysis and results in an easy-to-understand manner for patients and health professionals. I think more details are needed and how the proposed system and visualizations addressed the problems raised in the introduction. 

  1. The title seems too general to reflect the description of the main context, which focuses on a system improving diagnoses in SAS and OD.
  2. In the abstract, a general summary of current problems in Japan seems disconnected from the rest, which failed to provide concrete reasoning on why the system is important and needed for SAS and OD in children. 
  3. In lIne 111, I disagreed with the conclusion that machine learning and AI are difficult to visualize the analytical process. 
  4. In line 172, it mentioned using a simple method of multivariate analysis and the results are promising. However, no description of how the multivariate analysis was conducted nor results were shown. 
  5. Following the last bullet point, no data description was provided on the patients used in the analysis in line 172 and 307. 
  6. In line 89, the author said that the data are insufficient for machine learning. However, in line 174, the author contradicted it by saying that they developing analytical methods and new devices that combine the above methods using machine learning. It is not clear how machine learning is used throughout the analysis. If used, please describe the exact method rather than quoting it as "machine learning". 
  7. In line 307, is it "correlation coefficient" rather than "intercorrelation coefficient"? 
  8. Regarding conclusions drawn based on Figure 5, it is notable that the position map indicates the region of correlation coefficients for health patients. But it is not clear how far the points are away from the boundary will be diagnosed as OD patients. 
  9. In Section 6, Medical Support System, the authors stated the fact that the underlying algorithm and results interpretation can be obscure in analyses. Firstly, there is no detailed description of how this proposed medical support system works. Then, I don't see how it can improve this situation. Did the authors mean that it provides a simple visual verification on FFT? 
  10. Given that author mentioned addressing several problems (declining birthrate, aging population) existing in Japan, it is not clear how this new system will be generalized to a much broader setting. 

Author Response

Thank you very much for your comments and advice. I have made some additions and corrections, please check them.

This paper presents two visualization examples and one medical support system based on a device to diagnose SAS and OD in children cohorts. The goal is to provide a system providing data analysis and results in an easy-to-understand manner for patients and health professionals. I think more details are needed and how the proposed system and visualizations addressed the problems raised in the introduction. I followed your 10 comments and advice and added more detailed information rice field. Also, the title has been changed to symbolize the content of the dissertation.

1. The title seems too general to reflect the description of the main context, which focuses on a system improving diagnoses in SAS and OD.

    ->  I changed it.

2. In the abstract, a general summary of current problems in Japan seems disconnected from the rest, which failed to provide concrete reasoning on why the system is important and needed for SAS and OD in children. 

    -> In line 27: In the introduction I have added a few references.

3. In line 111, I disagreed with the conclusion that machine learning and AI are difficult to visualize the analytical process. 

    -> In line 198: I have added.

4. In line 172, it mentioned using a simple method of multivariate analysis and the results are promising. However, no description of how the multivariate analysis was conducted nor results were shown. 

    -> line 198: I have added.

5. Following the last bullet point, no data description was provided on the patients used in the analysis in line 172 and 307. 

    -> In line 200 and 335, I added.

6. In line 89, the author said that the data are insufficient for machine learning. However, in line 174, the author contradicted it by saying that they developing analytical methods and new devices that combine the above methods using machine learning. It is not clear how machine learning is used throughout the analysis. If used, please describe the exact method rather than quoting it as "machine learning". 

    -> From line 196, I added an explanation.

7. In line 307, is it "correlation coefficient" rather than "intercorrelation coefficient"? 

    -> In line 329, I changed it.

8. Regarding conclusions drawn based on Figure 5, it is notable that the position map indicates the region of correlation coefficients for health patients. But it is not clear how far the points are away from the boundary will be diagnosed as OD patients. 

    -> From line 341, I added the explanation about it.

9. In Section 6, Medical Support System, the authors stated the fact that the underlying algorithm and results interpretation can be obscure in analyses. Firstly, there is no detailed description of how this proposed medical support system works. Then, I don't see how it can improve this situation. Did the authors mean that it provides a simple visual verification on FFT? 

    -> From line 394, I added the explanation.

10. Given that author mentioned addressing several problems (declining birthrate, aging population) existing in Japan, it is not clear how this new system will be generalized to a much broader setting.

-> In line 435, I added it.

Reviewer 2 Report

1.    Please add the future study in the conclusion section.
2.    The introduction section lacks of exploring in the medical support system.
3.    The parameters’ setting of machine learning cannot be found  in this paper. 
4.    It could be discussed about the sensitivity analysis of the system’s performances based on different parameters’ setting. 

Author Response

Thank you very much for your comments and advice. I have made some additions and corrections, please check them.

1. Please add the future study in the conclusion section.

    -> From line 435, I added it.

2. The introduction section lacks of exploring in the medical support system.

    -> From line 30, I added it.

3.    The parameters’ setting of machine learning cannot be found in this paper. 

    -> In line 282, I added it.

4.    It could be discussed about the sensitivity analysis of the system’s performances based on different parameters’ setting. 

    -> In line 430, I added my plan about it.

Round 2

Reviewer 1 Report

Thank you for addressing my comments. Here are my replies. 

4. In line 172, it mentioned using a simple method of multivariate analysis and the results are promising. However, no description of how the multivariate analysis was conducted nor results were shown. 

    -> line 198: I have added.

  • I should have clarified what I've meant earlier by describing the method and results for the multivariate analysis. What variables were fitted to the model? To interpret the results, please report the coefficient and its associated p values for those variables of interest. 

5. Following the last bullet point, no data description was provided on the patients used in the analysis in line 172 and 307. 

    -> In line 200 and 335, I added.

  • I was expecting a more detailed description of the data (maybe one or two paragraphs) rather than simply listing the sample size (42 subjects and 35 subjects). For example, as for the database mentioned in Line 56, how many patients were recruited? Did they have completed the study with full follow-up? Are there any missing data in those subjects? What variables were collected on subjects? etc.

Author Response

Dear Reviewer1:

Following your advice, I modified the paper as follows:

*****
4. In line 172, it mentioned using a simple method of multivariate analysis and the results are promising. However, no description of how the multivariate analysis was conducted nor results were shown. 

    -> line 198: I have added.

I should have clarified what I've meant earlier by describing the method and results for the multivariate analysis. What variables were fitted to the model? To interpret the results, please report the coefficient and its associated p values for those variables of interest. 

    -> line 201: I have added.
*****
5. Following the last bullet point, no data description was provided on the patients used in the analysis in line 172 and 307. 

    -> In line 200 and 335, I added.

I was expecting a more detailed description of the data (maybe one or two paragraphs) rather than simply listing the sample size (42 subjects and 35 subjects). For example, as for the database mentioned in Line 56, how many patients were recruited? Did they have completed the study with full follow-up? Are there any missing data in those subjects? What variables were collected on subjects? etc.

    -> line 201 and 346: I have added.

Best Regards,
Miyori Shirasuna

Reviewer 2 Report

This paper could be published in this journal, Sensors.

Author Response

Dear reviewer
Thank you for your approval.

Best Regards,

Miyori Shirasuna